# Regional impact of aging population on economic development in China: Evidence from panel threshold regression (PTR)

**Yifan Liang**[1☯], **Nur Syazwani Mazlan**[1☯]*, **Azali Bin Mohamed**[1☯], **Nor Yasmin Binti Mhd Bani**[1☯], **Bufan Liang**[2☯]

**1** School of Business and Economics, Universiti Putra Malaysia, Selangor, Malaysia, **2** School of Economics, University of Bristol, Bristol, United Kingdom

☯ These authors contributed equally to this work.

* nur.syazwani@upm.edu.my

**Data Availability Statement:** All relevant data are within the manuscript and its Supporting Information files.

## Abstract

The aging population is a common problem faced by most countries in the world. This study uses 18 years (from 2002 to 2019) of panel data from 31 regions in China (excluding Hong Kong, Macao, and Taiwan Province), and establishes a panel threshold regression model to study the non-linear impact of the aging population on economic development. It is different from traditional research in that this paper divides 31 regions in China into three regions: Eastern, Central, and Western according to the classification standard of the National Bureau of Statistics of China and compares the different impacts of the aging population on economic development in the three regions. Although this study finds that the aging population promotes the economy of China's eastern, central, and western regions, different threshold variables have dramatically different influences. When the sum of export and import is the threshold variable, the impact of the aging population on the eastern and the central region of China is significantly larger than that of the western region of China. However, when the unemployment rate is the threshold variable, the impact of the aging population on the western region of China is dramatically higher than the other regions' impact. Thus, one of the contributions of this study is that if the local government wants to increase the positive impact of the aging population on the per capita GDP of China, the local governments of different regions should advocate more policies that align with their economic situation rather than always emulating policies from other regions.

## Background

The aging population is a new hot issue raised in recent decades. According to the World Health Organization's definition, the aging population refers to the number of elderly people over 65 years old accounting for more than 7% of the total population in a country or region. The aging population is a sociological issue, but Solow and Modigliani and Ando built a bridge between the aging population and economic development through their research and theories.

**Funding:** The author(s) received no specific funding for this work.

**Competing interests:** The authors have declared that no competing interests exist.

From the perspective of the labor force, Solow [1] pointed out that the population growth rate was one of the important factors affecting economic development. However, he assumed that the whole population of a country or region was an effective labor force, which was not in line with reality. Therefore, based on Solow's research, Modigliani and Ando [2] put forward the life cycle hypothesis. This hypothesis roughly divides a person's life into two periods: the working period (15–64 years old) and the retirement period (over 65 years old). Modigliani and Ando assumed that people lost their ability to work and had no income during the retirement period. Thus, all consumption of old people during the retirement period came from their savings from the work period. Therefore, as the population ages, a country's total savings will continue to decline and ultimately hinder economic growth.

Based on the above studies and theories, many scholars use linear models to study the impact of the aging population on economic growth, but this research method has major flaws. For example, Yang et al. [3] found that the aging population would inhibit economic growth by reducing the ecological footprint; Lukyanets et al. [4] also found that the aging population would reduce economic growth in Russia; Scott et al. [5] and Liu [6] got the same conclusion. However, Pham and Vo [7], Mamun *et al.* [8], and Wang *et al.* [9] found that the aging population would promote economic growth. The reason why this study believes that the conclusions of these studies are contradictory is that: in the early stage of the aging population, although the elderly population is increasing, the young population (15–64 years old) is also increasing. At this stage, because the increase in the number of the elderly population is less than the increase in the number of the young population, the negative impact of the elderly population on economic growth will be offset by the positive impact of the young population on economic growth. Therefore, in the early stage of the aging population, the aging population will promote economic growth. However, in the late stage of the aging population, because the increase in the number of the elderly population is greater than that of the young population, the aging population will inhibit economic growth. Therefore, if the linear model is used to study the impact of the aging population on economic growth, the linear model only can reflect partial information of data. Yang *et al.*, using the across-country data, proved there were nonlinear effects between the aging population and economic growth [10]. Therefore, this study will use the nonlinear model to study the impact of the aging population on economic growth.

In addition, most scholars currently study the impact of the aging population on economic growth in the context of developed countries, but this is inconsistent with the future trend of the global aging population. According to the World Health Organization (WHO) data, there will be 2.1 billion people over the age of 60 by 2050, and most older people will live in developing countries. Therefore, this study takes China as an example to study the impact of the aging population on the economic growth of developing countries.

China is a socialist country, and local governments must follow the policies of the central government. Thus, from the policy perspective, the impact of the aging population on all regions of China should be the same. However, it is not. There are significant differences in the relationship between the aging population and economic development in different regions of China (see S1 Appendix). For example, the aging population in Shanghai was negatively correlated with economic development from 2002 to 2011 but positively correlated with economic development thereafter. While the relationship between the aging population and economic development is firstly positively correlated and then negatively correlated in Heilongjiang province during the same period. This finding is the exact opposite of what happened in Shanghai. Besides, the aging population in Tibet Autonomous Region does not seem to be directly related to economic development. Apart from that, the aging population in Guangxi is almost positively related to economic development.

Besides, in studies comparing the impact of the aging population on economic growth in different regions, most scholars, like Yang *et al.*, compare the differences between countries [10]. However, different countries have different cultures, social systems, policies, and other factors. Therefore, it is difficult for different countries to learn from each other's experiences. Thus, based on the current data in China and the research perspective of scholars, this study proposes a new research question: Why does the aging population have different impacts on economic development in different regions of China? Because China has 31 regions (excluding Hong Kong, Macao, and Taiwan), comparing the differences between regions is unrealistic. Therefore, this study follows the geographical classification of Xie *et al.* [11] and divides China's 31 regions into three regions: eastern, central, and western. The study aims to examine the different impacts of the aging population on economic development in the three regions of China.

The contributions of this paper are as follows: first, this study provides experiences to developing countries on the impact of the aging population on economic growth; second, this study fills in the gaps in the impact of the aging population on regional differences in the same country. Third, this study provides policy suggestions for China's regions that are relatively backward in economic development. Fourth, this study uses a more advanced panel threshold regression (PTR), which can provide more comprehensive and accurate conclusions.

This study aims to examine the different impacts of the aging population on different regions in China. Besides, the rest of this study is structured as follows: Section 2 examines the relevant and recent studies and proposes hypotheses based on them; Section 3 explains the methodology and data processing and sources; Section 4 presents the empirical research and discussions, and the final section illustrates the empirical results and policy recommendations to the Chinese government.

## Literature review

Most scholars take developed countries as their research objects when they study the impact of population aging on economic growth. For example, Maestas *et al.* [12] found that the aging population inhibits U.S. economic growth. The main reason is a decline in labor productivity. The second reason is a decline in the number of workers. This study concluded that for every 10% increase in the population over 60 years old, U.S. GDP fell by 5.5%. However, it is not realistic to define people over 60 years old as elderly. Because of the development of the medical level, the average life expectancy has greatly increased. Therefore, the Chinese government formulated a delayed retirement plan in 2015 and officially implemented it in 2022. Gordon [13] compared the panel data of the United States and other developed countries and came to the same conclusion. Besides, they found that the unemployment and gap between the rich and the poor would also impede economic development. Nevertheless, the comparison between countries cannot provide their own experience because different countries have different political systems, cultural backgrounds, development stages, etc. are not the same.

In the study of developed countries or regions in East Asia, Goh and McNown [14] and Huang *et al.* [15] both used time series methods. The former finds that the aging population inhibits Japan's economic growth due to the decline in total savings, which is consistent with the conclusion of Modigliani and Ando [2]. The latter finds that the impact of the aging population on the economic growth of the Taiwan province of China is more complex. Specifically, the aging population would restrain economic growth. However, the aging population would increase total factor productivity (TFP) at the same time. Although the two studies explain why the aging population inhibits economic growth from different perspectives, the time series data reflect limited information. Both Lee *et al.* [16] and Kim and Lee [17] found that the

aging population would depress South Korea's economic growth. However, the variables used by Lee *et al.* [16] cannot accurately measure the aging population. In their study, the variables measuring the aging population are the proportion of the young population and the proportion of the old population. At present, a more accurate variable is the old dependency ratio. Kim and Lee [17] assumed that the age distribution was constant in their research. Nevertheless, the aging population is caused by the uneven age distribution caused by the continuous decline in the birth rate. In addition, they did not consider the role of capital deepening.

In addition, some scholars have studied the impact of the aging population on the economic growth of European countries. For example, Kamiguchi and Tamia [18] use an intergenerational iterative model to find that the aging population inhibits economic growth by increasing public debt. Jackson [19] came to the same conclusion and found that the decline in innovation ability is also one of the reasons why the aging population inhibits economic growth. In addition, Lorenz *et al.* [20] also found that the healthcare expenditure caused by the aging population is the main reason for reducing economic growth. These three studies inspired this study to consider the impact of changes in social welfare on economic growth. Because unlike developed countries, China's per capita GDP when the population age is far lower than developed countries' per capita GDP when the population age. Therefore, the Chinese government cannot ignore the public debt caused by the aging population. In addition, Cooley *et al.* [21], Pascual-Saez *et al.* [22], Šídlo *et al.* [23], and Bobeica *et al.* [24] have confirmed that the aging population would cause total savings to decrease, inflation and other things that would hold back the economy. Through the above literature, it can be found that the impact of the aging population on developed countries or regions is negative.

A few scholars surprisingly found that the aging population would promote economic growth in developing countries. For example, Mamun *et al.* [8] found that the aging population boosts economic growth in Bangladesh. Their study supports Prettner 's conclusions, and they believe that the reason for this phenomenon is that the Bangladeshi government does not provide enough subsidies to the elderly [25]. However, their research lacks human capital data. Ifa and Guetat studied data from Tunisia and Morocco and found that the aging population would force the government to invest more capital in health and ultimately promote economic growth [26]. Research by Wang *et al.* [27] shows that life expectancy and economic development were positively correlated. This conclusion proves the research of Ifa and Guetat [26]. However, the study did not consider the increased retirement age due to increases in life expectancy. Thus, we do not know how an increase in life expectancy stimulates economic growth.

However, the aging population does not affect developing countries uniformly. For example, Peña [28] found that the increase in public debt caused by the aging population was the main reason for El Salvador's economic growth rate slowdown. This finding is consistent with research conclusions in developed countries. In addition, Miri *et al.* [29] found through data from Iran that the relationship between the aging population and economic growth was nonlinear. But they did not use the old dependency ratio but the proportion of the elderly population and the proportion of the youth population.

The above studies show that the impact of the aging population on the economic growth of developed countries is quite different from the impact of the aging population on that of developing countries. The possible reasons for this difference are explained in the Introduction section of this study. Therefore, this study proposes the following assumptions:

H1: The aging population has a significant impact on economic growth in developing
    countries

H1o: The relationship between the aging population and economic development in developing countries is linear

H1a: The relationship between the aging population and economic development in developing countries is nonlinear.

Some scholars use cross-country data to study the impact of the aging population on economic growth. For example, Vollset *et al*., based on data from 195 countries, found that the aging population and increased public debt lower GDP per capita [30]. However, some immigration data are missing in the research. Immigration means that a country's labor force sharply increases, which will significantly alleviate the negative impact of the aging population on economic growth. Therefore, the absence of some immigration data will affect the accuracy of the conclusion. De Albuquerque *et al*. [31] proved an inverted U-shaped relationship between the aging population and economic growth from an inflation perspective. Specifically, an increase in the number of older people is inflationary, and an increase in young people is deflationary. However, the study does not fit China's national conditions. In traditional Chinese culture, the elderly will not consume all their savings before dying. In other words, they are more inclined to endow their children. Eggertsson *et al*. [32] examine the impact of the aging population on economic growth in OECD member countries. Studies have shown that during 1990–2008, the aging population promoted economic growth; during 2008–2015, the aging population inhibited economic growth. However, the economic level of developed countries facing an aging population is entirely different from that of developing countries'. In addition, Lee and Shin [33] and Yang *et al*. [10] also found an inverted U-shaped relationship between the aging population and economic growth. The above studies using cross-country data all have an obvious limitation: they do not compare and research different countries or regions.

In examining the impact of the aging population on China's economic growth, Chen *et al*. [34] found that the decline in the labor force participation rate caused by population aging was the main reason for the slowdown in economic growth. However, they used time series data and did not compare the differences between different provinces or regions within China. Liu's research shows that the aging population would inhibit China's economic development [6]. In this study, the author compared the changes in China's economic development before and after the implementation of the two-child policy. Research by Bai and Lei [35] shows that the aging population has hurt China's economic growth since 2000. On the one hand, the aging population reduced labor productivity and labor participation rates. On the other hand, the aging population declined the total saving of society. Wang and Yu [9] proved that the aging population would inhibit economic growth from the perspective of consumption. Cao *et al*. [36] proposed that China should reduce the negative impact of the aging population on economic growth by increasing education, expanding the urbanization rate, and postponing retirement. However, in many studies on the impact of the aging population on China's economic growth, scholars only analyze the effects of the aging population on economic growth from a macro perspective without comparing and researching the effects of the aging population on different regions of China. Therefore, this study puts forward the following hypotheses based on the above research:

H2o: The aging population will have differential impacts on economic growth in different regions of China.

H2a: The aging population will not have differential impacts on economic growth in different regions of China.

## Methodology

### Data sources and processing

This study's original data are from the National Bureau of Statistics of China. To make the data more stable and avoid heteroscedasticity, this study takes the logarithm of all the data. Apart from that, this study uses panel data which includes 31 regions of China (Hong Kong, Macao, and Taiwan Province excluded), and the time frame is 18 years (2002 to 2019). The definition and calculation of variables can be seen in Table 1 below. Specifically, explained variable is per capita GDP. The explanatory variable is the old-dependency ratio. The threshold variables are the unemployment rate and the sum of exports and imports. Lastly, the control variables are the child-dependent ratio, the percentage of working people, and the illiteracy rate.

Because China has 31 regions (excluding Hong Kong, Macao, and Taiwan), it is unrealistic to pairwise compare the differences between regions. Thus, scholars usually divided China into several regions to compare and research. However, different scholars adopt different classification methods. For example, Zou *et al.* [37] and Wang *et al.* [27] directly divide China into rural and urban. However, in the past 20 years, China's urbanization rate has increased rapidly. Therefore, this method is no longer suitable. In addition, Liu *et al.* [38] only studied Northwest China. It cannot comprehensively reflect the impact of the aging population of China. Thus, this study divides 31 regions of China into three regions: the eastern, central, and western regions. The eastern region has eleven regions, the central region has eight regions, and the western region has twelve regions. The classification of this study is based on the research of Xie *et al.* [11] and the National Bureau of Statistics of China which is in Table 2 below.

### Model specification

This study will establish a panel threshold regression model. PTR model views a threshold variable as an unknown variable, establishes a piecewise function, and examines and estimates the corresponding threshold values and the effects. Lee and Shin [33] confirmed the non-linear relationship between the aging population and economic development. Hence, this study constructs the threshold model based on the research of Hansen [39] to study the relationship between the aging population and economic development at different intervals. In this study, a single threshold regression model will be constructed first and then extended to a multiple-threshold regression model. The construction of the single threshold regression model is as follows:

$$Y_{it} = \delta_i + \beta' X_{it} + \theta_1 g_{it} I(d_{it} \leq \gamma) + \theta_2 g_{it} I(d_{it} > \gamma) + \omega_{it} \qquad (1)$$

**Table 1. Definition of research variables.**

| Abbreviation | Variable Name | Calculation Method |
|:---:|:---:|:---:|
| LPCGDP | Log per capita GDP | $\frac{Total\ GDP}{resident\ population}$ |
| LODR | Log old-dependency ratio | $\frac{over\ 65\ years\ old\ people}{15\ to\ 64\ years\ old\ people}$ |
| LEI | Log sum of export and import | $export\ value + import\ value$ |
| LUE | Log unemployment rate | $\frac{the\ number\ of\ unemployment}{hte\ number\ of\ total\ people}$ |
| LCDR | Log child-dependency ratio | $\frac{the\ number\ of\ childern}{15\ to\ 64\ years\ old\ people}$ |
| LWP | Log the number of working people | $\frac{15\ to\ 64\ years\ old\ people}{hte\ number\ of\ total\ people}$ |
| LIL | Log illiteracy rate | $\frac{the\ number\ of\ illiteracy}{hte\ number\ of\ total\ people}$ |

**Table 2. Classification of 31 regions of China.**

| Eastern Region of China | Central Region of China | Western Region of China |
|---|---|---|
| Beijing | Shanxi | Inner Mongolia |
| Tianjin | Jilin | Guangxi |
| Hebei | Heilongjiang | Chongqing |
| Liaoning | Anhui | Sichuan |
| Shanghai | Jiangxi | Guizhou |
| Jiangsu | Hunan | Yunnan |
| Zhejiang | Hubei | Shaanxi |
| Fujian | Henan | Gansu |
| Shandong | | Qinghai |
| Guangdong | | Ningxia |
| Hainan | | Xinjiang |
| | | Tibet |

where i represents the three regions of China; t represents the time; $Y_{it}$ represents the dependent variable which is per capita GDP; $g_{it}$ represents the independent variable which is the old dependency ratio; $X_{it}$ represents control variables which are the percentage of working people, the children dependency ratio and the illiteracy rate;

$\beta'$ is the co-efficient of variables;

$\gamma$ represents the specific value of threshold; $d_{it}$ represents threshold variables which are the sum of export and import, and the unemployment rate; and I(.) represents an indicator function.

$\delta_i$ represents an unpredictable factor and it has no relationship with the unit.

$\omega_{it}$ is an error term and this study assumes $\omega_{it}$ is independent and identically distributed.

After eliminating the individual effects in each group from each observation by subtracting the average value, Eq (1) can be written as follow:

$$Y_{it}^* = \beta' X_{it}^* + \theta_1 g_{it}^* I(d_{it} \leq \gamma) + \theta_2 g_{it}^* I(d_{it} > \gamma) + \omega_{it}^* \tag{2}$$

After stacking the observations, Eq (2) can be written as follow:

$$Y^* = X^*(\gamma)\theta + \omega^* \tag{3}$$

Usually, people estimate Eq (3) via ordinary least square to acquire the value of $\theta$, and the function is as follows:

$$\theta(\gamma) = \frac{X^*(\gamma)' Y^*}{X^*(\gamma)' X^*(\gamma)} \tag{4}$$

The function of the sum of the squared residuals is as follows:

$$S_1(\gamma) = \hat{e}^*(\gamma)' \hat{e}^*(\gamma) \tag{5}$$

Besides, the function of the residual vector is as follows:

$$\hat{e}^*(\gamma) = Y^* - X^*(\gamma)\theta(\gamma) \tag{6}$$

After the parameter is obtained, there are two steps to estimate the threshold proposed by Huang *et al*. [40]. The first step is to test whether the model has a threshold effect. Hence, the null hypothesis is that there is a linear relationship between variables and the alternative

hypothesis is that a threshold effect exists. The function is as follows:

$$F_1 = \frac{S_0 - S_1(\hat{\gamma})}{\hat{\sigma}^2} \tag{7}$$

In the null hypothesis, the specific value of $\gamma$ cannot be calculated. Apart from that, the distribution of the $F_1$ statistic is non-standard either. According to the suggestion of Hansen (2000), people can estimate progressive distribution and P-values through bootstrap.

The second step is to test whether the estimated threshold value equals the actual value. Therefore, the null hypothesis of the test is that the outcomes are the same, and the alternative hypothesis of the test is that the outcomes are not the same. The function of likelihood ratio statistic is as follows:

$$LR_1(\gamma) = \frac{S_1(\hat{\gamma}) - S_1(\gamma)}{\hat{\gamma}^2} \tag{8}$$

Due to the distribution being non-standard, when using $LR_1(\gamma_0) \leq -2\ln(1 - \sqrt{1-\alpha})$, we still cannot reject the null hypothesis while $\alpha$ is significant.

Apart from the single threshold model, there is also a multi-threshold model in empirical analysis. Therefore, this study takes the double threshold as an example to illustrate the threshold regression. The function is as follows:

$$Y_{it} = \delta_i + \beta' X_{it} + \theta_1 g_{it} I(d_{it} \leq \gamma_1) + \theta_2 g_{it} I(\gamma_1 < d_{it} \leq \gamma_2) + \theta_3 g_{it} I(d_{it} > \gamma_2) + \omega_{it} \tag{9}$$

In double threshold regression, we assume that we already have estimated the specific value of the $\hat{\gamma}_1$. Hence, we only need to estimate $\hat{\gamma}_2$ The function is as follows:

$$S_2^{\alpha} = \begin{cases} S(\hat{\gamma}_1, \gamma_2) \ if \hat{\gamma}_1 < \gamma_2 \\ S(\gamma_2, \hat{\gamma}_1) \ if \hat{\gamma}_1 > \gamma_2 \end{cases} \tag{10}$$

$$\hat{\gamma}_2^{\alpha} = argmin S_2^{\alpha}(\gamma_2) \tag{11}$$

where

$\hat{\gamma}_2^{\alpha}$ represents progressively effective and $\gamma_2^{\alpha}$ can be fixed for $\hat{\gamma}_1$, and then $\dfrac{r}{2}$ can be recalculated to acquire a consistent value.

## Unit root test

All econometric methods including time series must first pass the unit root test between variables because the variables cannot pass the unit root test This means that there may be spurious regressions between the variables, and the result will be meaningless.

In more literature reviews, for example, Pesaran [41] pointed out that the cross-sectional dependence problem exists in panel data. The reason for this problem is the unobserved components and common shocks. Therefore, cross-sectional dependence issues reduce the accuracy of the results and can even lead to false results.

The panel data is composed of cross-sectional and time-series data. Therefore, the stationarity of the data needs to be considered in the diagnostic test. However, the first generation of unit root tests did not consider the cross-sectional dependency problem. Specifically, if the panel data has a cross-sectional dependency problem, the first-generation unit root test will erroneously reject the null hypothesis.

Although the second-generation unit root test considers the cross-sectional dependency problem, Urbain and Westerlund [42] believe that the homogeneity assumption in the cross-section dependency problem is still not true. Therefore, this study uses the unit root test of Pesaran [41]. The formula can be written as follow:

$$a_{it} = \delta_i + \beta_{it} b_{it} + \mu_{it} \tag{12}$$

In the Eq (12), $a_{it}$ is the dependent variable, $b_{it}$ is the independent variable, $\beta_{it}$ is the coefficient of the independent variable and $\delta_i$ is an individual nuisance parameter that will not change with time. For the subscript, i is unit and t is time.

## Cointegration test

Most macroeconomic variables are trended, but the trended time series may cause serious problems in empirical models due to spurious regressions. In general, one of the methods to solve the spurious regression is by differencing the series to achieve stationarity. Then we use the stationarity series and analyze the outcome. However, this method will produce other problems. For example, differencing all variables means that the error term is also differenced. This will produce a non-invertible moving average error process, which will lead to serious estimation problems. In addition, if we difference the variables, the model will not give us a unique long-run solution.

Engle and Granger [43] firstly pointed out that if there is a co-integration relationship between variables, the model will not have spurious regression problems. Next, more panel co-integration test methods are employed in the empirical model. For example, both Pedroni [44] and Persyn and Westerlund [45] proposed a panel co-integration test, but this study adopted the method of Persyn and Westerlund [45]. This method not only considers the structural breaks but also the lead-lag lengths when we employed short-duration data. The specific bootstrap panel co-integration formula is as follows:

$$\Delta Y_{it} = \delta' d_t + \alpha_i \left( Y_{i,t-1} - \beta_i' X_{i,t-1} \right) + \sum_{j=q_i}^{q_i} a_{ij} \Delta Y_{i,t-j} + \sum_{j=q_i}^{q_i} \gamma_{ij} \Delta X_{i,t-j} + e_{it} \tag{13}$$

In the above equation, $\Delta Y_{it}$ is an endogenous variable, and $d_t$ refers to the deterministic component. $i$ is unit and $t$ is time.

## Empirical result

To research the impact of the aging population on economic development, this study views the unemployment rate and the sum of imports and exports on economic development as the threshold variable. It uses 18 years of data (2002 to 2019) from 31 regions of China. During this process, the study divides the 31 regions of China into three groups: the eastern, central, and western regions to compare the different threshold effects of the aging population on economic development in the three regions.

## Descriptive statistics summary

Firstly, to make the data more stable and avoid heteroscedasticity, this study takes the logarithm of all the data. The lpcgdp is log the per capita GDP; the lodr is log the old-dependency ratio; the lil is log the illiteracy; the lei is log the sum of export and import; lue is log the unemployment rate; lcdr is log the child-dependency ratio; and lwp is log the number of working people.

Tables 3–8 list the descriptive statistics and pairwise correlation tests between variables of the eastern, central, and western regions of China. The mean value of lpcgdp in the three regions of China is almost the same. However, the standard deviations of lpcgdp of the eastern and central regions of China are dramatically higher than that of the western region of China. It means that the income gap between poor and rich people in the eastern and central regions of China is significantly higher than that in the western region of China. In terms of the sum of export and import, different cities in the eastern and western regions of China have huge gaps. However, the gap between the cities in the central region of China is relatively small. Besides, people in the eastern and central regions of China received a better education than people in the western region of China.

Secondly, this study processes unit root test to make sure that all data are stable. The outcomes from Tables 9–11 show that all data are significant in the first difference.

Thirdly, this study processes the cointegration test to make sure that there is no spurious regression. In the co-integration test, this study utilizes the first difference of variables, and the null hypothesis of the co-integration test is that there is no co-integration. The outcomes from Table 12 show no spurious regression in China's eastern, central, and western regions.

Thirdly, this study tests whether there is a threshold effect between variables. In this test, this study utilizes Stata 15.1 version and sets bootstrap to 300, grid to 400, and trim to 0.01. In Table 13, model 1, model 2, and model 3 respectively represent the result of the eastern, central, and western regions of China when the LEI is the threshold variable. The outcomes show that in the eastern region and the western region of China, there is only one threshold value, respectively. However, in the central region of China, there are two threshold values.

In Table 14, model 4, model 5, and model 6 respectively represent the result of the eastern, central, and western regions of China when the LUE is the threshold variable. The outcomes show that in the eastern and central regions of China, there is only one threshold value, separately. However, there are two threshold values in the western region of China.

## Results and discussion

### Sum of exports and imports

After the data pass all pre-requisite tests, this study process threshold regression and outcomes are presented in Table 15. Model 1, model 2, and model 3 represent the results of the eastern, central, and western regions of China when the LEI is the threshold variable. The outcomes show that the threshold value of LEI in the eastern region of China is 18.80. Therefore, due to all variables taking the logarithm, the actual value of the sum of exports and imports is 14.613 billion dollars. Similarly, the data of the western region of China have been divided into two groups by the level of LEI and the threshold value of LEI is 13.1 (0.049 billion dollars).

**Table 3. Descriptive statistics of eastern region of China.**

| Variable | Obs | Mean | Std. Dev | Min | Max |
|---|---|---|---|---|---|
| Lpcgdp | 198 | 10.644 | 0.688 | 8.992 | 12.011 |
| Lord | 198 | 2.604 | 0.203 | 2.152 | 3.17 |
| Lil | 198 | 1.545 | 0.585 | 0.207 | 2.725 |
| Lei | 198 | 18.594 | 1.324 | 14.399 | 20.971 |
| Lue | 198 | 1.154 | 0.316 | 0.182 | 1.872 |
| Lcdr | 198 | 2.945 | 0.323 | 2.262 | 3.701 |
| Lwp | 198 | 4.315 | 0.045 | 4.181 | 4.429 |

**Table 4. Descriptive statistics of central region of China.**

| Variable | Obs | Mean | Std. Dev. | Min | Max |
|---|---|---|---|---|---|
| Lpcgdp | 144 | 10.007 | 0.657 | 8.671 | 11.248 |
| Lord | 144 | 2.555 | 0.192 | 2.116 | 3.03 |
| Lil | 144 | 1.636 | 0.531 | 0.577 | 2.957 |
| Lei | 144 | 16.679 | 0.796 | 14.507 | 18.293 |
| Lue | 144 | 1.283 | 0.153 | 0.875 | 1.589 |
| Lcdr | 144 | 3.16 | 0.272 | 2.549 | 3.632 |
| Lwp | 144 | 4.288 | 0.05 | 4.197 | 4.385 |

However, the data of the central region of China are divided into three groups and the threshold values of LEI separately are 16.321 (1.225 billion dollars) and 17.075 (2.604 billion dollars).

Firstly, Table 16 presents that the aging population has a significant positive relationship with per capita GDP in the three regions of China when the LEI is the threshold variable. Besides, for the eastern China and western regions of China, there is only one turning point for the impact of the aging population on economic growth. However, there are two turning points in central China.

For Eastern China, the turning point of the impact of the aging population on economic growth is when the LEI equals 18.80. In other words, when the LEI is less than or equal to 18.80, the coefficient of LDOR is 2.738; when the LEI is greater than 18.80, the coefficient of LODR is 2.872. Therefore, when the sum of imports and exports of eastern China is increased to more than 146.129 million yuan, the impact of the aging population on economic growth is optimal.

In terms of central China, when the LEI is less than or equal to 16.321, the coefficient of LODR is 3.050. When LEI exceeds the first turning point but does not exceed the second turning point, that is, when 16.321 is less than LEI and LEI is less than or equal to 17.075, the coefficient of LODR becomes 3.181. When the LEI crosses the second turning point, the coefficient of LODR is 3.290. Therefore, for central China, the government should increase the sum of imports and exports to more than 26.0365 million yuan because the impact of the aging population on economic growth is optimal at this LEI level.

For western China, LEI equal to 13.10 is a turning point for the impact of the aging population on economic growth. When the LEI is less than or equal to 13.10, the coefficient of LODR is 1.426; when the LEI is greater than 13.10, the coefficient of LODR is 1.667. Thus, from the perspective of promoting economic development, this study proposes that the government should increase the sum of imports and exports in western China to more than 489,000 yuan.

Regardless of whether it is eastern China, central China or western China, the aging population always promotes economic growth, and the marginal effect will gradually increase. But

**Table 5. Descriptive statistics of western region of China.**

| Variable | Obs | Mean | Std. Dev. | Min | Max |
|---|---|---|---|---|---|
| Lpcgdp | 216 | 9.92 | 0.73 | 8.089 | 11.236 |
| Lord | 216 | 2.468 | 0.258 | 1.902 | 3.144 |
| Lil | 216 | 2.276 | 0.667 | 1.006 | 4.005 |
| Lei | 216 | 15.6 | 1.524 | 11.738 | 18.464 |
| Lue | 216 | 1.278 | 0.174 | 0.742 | 1.589 |
| Lcdr | 216 | 3.333 | 0.212 | 2.803 | 3.8 |
| Lwp | 216 | 4.264 | 0.04 | 4.15 | 4.375 |

**Table 6. Pairwise correlation of eastern region of China.**

| Variables | (1) | (2) | (3) | (4) | (5) | (6) | (7) |
|---|---|---|---|---|---|---|---|
| (1) lpcgdp | 1.000 | | | | | | |
| (2) lodr | 0.328 | 1.000 | | | | | |
| (3) lil | -0.631 | -0.040 | 1.000 | | | | |
| (4) lei | 0.656 | 0.211 | -0.252 | 1.000 | | | |
| (5) lue | -0.402 | 0.233 | 0.376 | -0.087 | 1.000 | | |
| (6) lcdr | -0.588 | -0.328 | 0.597 | -0.335 | 0.134 | 1.000 | |
| (7) lwp | 0.464 | -0.125 | -0.573 | 0.280 | -0.231 | -0.881 | 1.000 |

the absolute values of the optimal points in the three regions have very large differences. Therefore, this study argues that more attention should be paid to the sum of imports and exports in the eastern region for the Chinese government. Thus, this study believes that China is currently in the early stage of the aging population.

Secondly, the impact of the unemployment rate on the per capita GDP is not the same. In China's western region, there is a significant negative linear relationship between the unemployment rate and the per capita GDP. This outcome is consistent with the research of Karikari-Apau and Abeti [46] and Egbejule and Oshiokpekhai [47]. Besides, Gordon [13] also found that the unemployment rate would decrease the birth rate. But in the eastern and central regions of China, there is no linear relationship between the unemployment rate and the per capita GDP. Thus, the government can increase the per capita GDP by decreasing the unemployment rate in the western region of China.

Thirdly, the child dependency ratio has different influences on the per capita GDP in the three regions of China. The relationship between the child dependency ratio and the per capita GDP is significantly positive in both the eastern and central regions of China. İmrohoroğlu and Zhao [48] found that the one-child policy increased the savings rate by reducing family insurance, which might be the reason why the child dependency ratio promoted economic growth. However, in terms of China's western region, the outcome points out that there is no linear relationship between the child dependency ratio and economic development. Apart from that, the degree of influence from the child dependency ratio in the eastern and central regions of China is almost the same, which are 2.418% and 2.758%, respectively. Research of Cruz and Ahmed [49] shows that an increase in the child dependency ratio decreased per capita GDP. They believed that if a family had few children, the family would have more per capita resources to consume or invest. Thus, a rise in the saving rate caused by an increase in the number of children is offset by a fall in consumption and investment. This may be the reason why there is no linear relationship between the child dependency ratio and economic growth in western China.

**Table 7. Pairwise correlation of central region of China.**

| Variables | (1) | (2) | (3) | (4) | (5) | (6) | (7) |
|---|---|---|---|---|---|---|---|
| (1) lpcgdp | 1.000 | | | | | | |
| (2) lodr | 0.629 | 1.000 | | | | | |
| (3) lil | -0.590 | 0.006 | 1.000 | | | | |
| (4) lei | 0.916 | 0.586 | -0.393 | 1.000 | | | |
| (5) lue | -0.471 | -0.297 | 0.143 | -0.467 | 1.000 | | |
| (6) lcdr | -0.281 | 0.075 | 0.601 | -0.126 | -0.320 | 1.000 | |
| (7) lwp | 0.045 | -0.431 | -0.543 | -0.076 | 0.399 | -0.926 | 1.000 |

**Table 8. Pairwise correlation of western region of China.**

| Variables | (1) | (2) | (3) | (4) | (5) | (6) | (7) |
|---|---|---|---|---|---|---|---|
| (1) lpcgdp | 1.000 | | | | | | |
| (2) lodr | 0.333 | 1.000 | | | | | |
| (3) lil | -0.568 | -0.481 | 1.000 | | | | |
| (4) lei | 0.526 | 0.666 | -0.832 | 1.000 | | | |
| (5) lue | -0.481 | -0.040 | 0.198 | -0.167 | 1.000 | | |
| (6) lcdr | -0.589 | -0.366 | 0.522 | -0.498 | 0.061 | 1.000 | |
| (7) lwp | 0.413 | -0.194 | -0.287 | 0.142 | -0.048 | -0.832 | 1.000 |

Fourthly, the influence of the percentage of working people is the same as the influence of the child dependency ratio. The degree of influence from the percentage of working people, however, is bigger. For example, in the eastern region of China, the coefficient of the percentage of working people is 17.487. It means that every 1% change in the percentage of working people will change 17.487% in the per capita GDP. Similarly, in terms of the central region of China, the degree of influence is 17.201%. Oliinyk's research points out that high-skilled immigration could significantly promote economic growth [50]. At the same time, immigration can increase the proportion of the working population in a short period. Therefore, increasing the percentage of working people is more effective than decreasing the illiteracy rate, decreasing the unemployment rate, increasing the child dependency ratio, and increasing the sum of export and import.

## Unemployment rate

In Table 17, model 4, model 5, and model 6 respectively represent the result of the eastern region, the central region, and the western region of China when the LUE is the threshold variable. The outcomes illustrate that the threshold value of LUE in the eastern region of China is 1.090. Due to all variables taking the logarithm, the actual value of the unemployment rate is 3.001%. Similarly, the data of the central region of China have been divided into two groups as well by the level of LUE and the threshold value of LUE is 1.361 (3.900%). However, the data of the western region of China have been divided into three groups and the threshold values of LUE separately are 1.131 (3.099%) and 1.308 (3.699%).

**Table 9. CIPS unit root test of eastern region of China.**

| Variables | Level | First Difference |
|---|---|---|
| LPCGDP | -2.705* | -3.280*** |
| LODR | -3.087*** | -4.715*** |
| LIL | -2.727 * | -4.526 *** |
| LEI | -2.558 | -3.270*** |
| LUE | -2.210 | -3.279 *** |
| LWP | -2.789** | -4.330*** |
| LCDR | -2.893** | 4.236*** |
| 1% | -3 | -2.45 |
| 5% | -2.77 | -2.22 |
| 10% | -2.65 | -2.11 |

Note: ***, **, and * show the levels of significance at 1%, 5%, and 10% respectively

**Table 10. CIPS unit root test of central region of China.**

| Variables | Level | First Difference |
|---|---|---|
| LPCGDP | -2.887* | -3.241*** |
| LODR | -2.906* | -4.345*** |
| LIL | -3.213** | -4.734*** |
| LEI | -2.100 | -3.781*** |
| LUE | -2.845* | -3.807*** |
| LCDR | -2.340 | -3.813*** |
| LWP | -1.935 | -3.704*** |
| 1% | -3.46 | -2.64 |
| 5% | -3.02 | -2.33 |
| 10% | -2.82 | -2.18 |

Note: ***, **, and * show the levels of significance at 1%, 5%, and 10% respectively

Firstly, Table 18 shows that the impact of the aging population on the per capita GDP is significantly positive in the three regions of China when the LUE is the threshold variable. However, unlike the sum of imports and exports, there is only one turning point in eastern and central China, but there are two turning points in western China.

For eastern China, when the LUE is less than or equal to 1.099, the coefficient of LODR is 1.582; when the LUE is greater than 1.099, the coefficient of LODR is 1.492, and both are statistically significant at the 1% level. Therefore, reducing the unemployment rate below 3.001% is more conducive to promoting the positive impact of the aging population on economic growth.

In terms of central China, the turning point of the impact of the aging population on economic growth is when the LUE is equal to 1.361. When the LUE is on the left side of the turning point, the coefficient of LODR is 1.267; when the LUE is on the right side of the turning point, the coefficient of LODR is 1.215, and both are significant at the 1% level. Thus, reducing the unemployment rate below 3.900% is more conducive to promoting the positive impact of population aging on economic growth.

For western China, there are two turning points for the impact of the aging population on economic growth: 1.131 and 1.308. When the LUE is less than or equal to 1.131, the coefficient of LODR is 1.907; when the LUE is between the two turning points, the coefficient of LODR is

**Table 11. CIPS unit root test of western region of China.**

| Variables | Level | First Difference |
|---|---|---|
| LPCGDP | -2.019 | -3.224*** |
| LODR | -3.597*** | -5.038*** |
| LIL | -2.856** | -4.455*** |
| LEI | -2.223 | -3.815*** |
| LUE | -1.433 | -3.196*** |
| LCDR | -2.727* | -4.004*** |
| LWP | -2.504 | -3.994*** |
| 1% | -3 | -2.45 |
| 5% | -2.77 | -2.22 |
| 10% | -2.65 | -2.11 |

Note: ***, **, and * show the levels of significance at 1%, 5%, and 10% respectively

**Table 12. Co-integration test of eastern, central, and western regions of China.**

| Region | Statistic | P-value |
|---|---|---|
| Eastern Variance Ratio | 6.1187 | 0.0000 |
| Central Variance Ratio | 12.6976 | 0.0000 |
| Western Variance Ratio | 10.0994 | 0.0000 |

1.787; when the LUE is greater than 1.308, the coefficient of LODR is 1.684. Therefore, from the perspective of promoting economic growth, the government should control the unemployment rate below 3.099%.

Reducing the unemployment rate will promote the positive effect of the aging population on economic growth in all three regions of China. However, the impact on the central region of China is relatively small, and the impact on the western region of China is relatively large. This study believes that the reason for this phenomenon is that the population in central China is relatively large, so the impact of changes in the unemployment rate is relatively small. However, the population in western China is smaller, so once the unemployment rate changes, the working population changes significantly, which ultimately affects economic growth.

Secondly, the degree of influence of LIL is almost the same in the central region and the western region of China. However, the LIL has much more influence in the eastern region of China. In other words, in the same situation, increasing the education opportunity will be more effective in the eastern region of China than in the other two regions. This conclusion is the same as the research of Chen *et al.* [34].

Thirdly, the degree of influence of LEI is not the same in the three regions of China. For the eastern and central regions of China, increasing the sum of export and import can significantly increase per capita GDP. On the other hand, for the western region of China, the sum of export and import only has limited benefits.

Fourthly, the LCDR only has a positive linear relationship with the per capita GDP in the eastern region of China. In the rest of the two regions, the outcome points out that there is no linear relationship. Similarly, LWP is almost the same as the LCDR, but the only difference is that LWP has much more influence than the LCDR.

## Conclusion remarks and implication

### Conclusions

Firstly, both the perspectives of the LEI and the LUE prove that the impact of the aging population on the per capita GDP in China is significantly positive. In other words, this study suggests that China is in the early stage of the aging population. Due to the number of working people and the number of older people increasing together, and the increasing speed of working people being faster than that of older people, the relationship between the aging population and the per capita GDP is positive. Although this conclusion is not the same as the previous

**Table 13. Threshold effect existence test when LEI is threshold variable.**

| | Threshold Test | RSS | MSE | F-stat | P-value | Critical Value | | |
|---|---|---|---|---|---|---|---|---|
| | | | | | | 10% | 5% | 1% |
| Model 1 | Single | 4.9763 | 0.0276 | 63.73 | 0.0000 | 17.9813 | 20.6128 | 25.8424 |
| Model 2 | Single | 3.4139 | 0.0271 | 36.00 | 0.0033 | 18.6442 | 21.7007 | 32.2779 |
| | Double | 2.5451 | 0.0202 | 43.01 | 0.0000 | 17.2788 | 20.9223 | 27.5555 |
| Model 3 | Single | 8.2405 | 0.0416 | 44.24 | 0.00067 | 25.4934 | 32.4275 | 39.4542 |

**Table 14. Threshold effect existence test when LUE is threshold variable.**

|  | Threshold Test | RSS | MSE | F-stat | P-value | Critical Value | | |
|---|---|---|---|---|---|---|---|---|
|  |  |  |  |  |  | 10% | 5% | 1% |
| Model 4 | Single | 3.3025 | 0.0183 | 35.59 | 0.0667 | 30.8858 | 40.0326 | 63.8884 |
| Model 5 | Single | 1.9819 | 0.0157 | 13.67 | 0.0700 | 11.3000 | 15.1975 | 23.4534 |
| Model 6 | Single | 8.5445 | 0.0432 | 58.83 | 0.0000 | 25.7488 | 32.2481 | 49.7251 |
|  | Double | 7.5750 | 0.0383 | 25.34 | 0.0100 | 17.9326 | 21.3436 | 24.9851 |

research, it still agrees with some authors' opinions. For example, Lee and Shin [33] found that the relationship between the aging population and economic development is non-linear. Besides, Mamun *et al.* [8] report that in Bangladesh, the aging population has a significant, long-run, positive relationship with the per capita GDP growth. Apart from that, Li and Zhang [51] found that in China, the aging population will have upward pressure on productivity which ultimately promotes economic development in China.

Secondly, this study innovative proves that under the aging population background, nowadays, there is a diminishing marginal effect between the sum of export and import and economic development in the central region of China as well as between the unemployment rate and economic development in the western region of China. For instance, in the central region of China, when the investment of the sum of export and import is over the first threshold value (1.225 billion dollars), the coefficient of LODR changes from 3.050 to 3.181: the gap is 0.131. However, when the investment of the sum of export and import is over the second threshold value (2.604 billion dollars), the coefficient of LODR changes from 3.050 to 3.181: the gap is 0.103. The benefits from the investment of the sum of export and import are decreasing. Similarly, the benefits from the unemployment rate are decreasing as well and the change is from 0.120 to 0.103.

Thirdly, from the perspectives of the LEI and the LUE, increasing education opportunities can promote significant economic development in all regions of China. Yet, education is more important in the eastern region of China than in the other two regions. The reason is that eight of the top ten regions with the highest per capita GDP are in the eastern region of China. In other words, most enterprises build their headquarters in the eastern region of China. Thus, the eastern region of China needs more educated workers than the other two regions. It explains why the coefficient of LIL in the eastern region of China is far lower than in the other two regions.

Fourthly, the most effective method to improve economic development under the aging population background in the three regions of China is increasing the percentage of working people. According to Tables 16 and 18, this study finds that the coefficients of LWP are around eight times that of other coefficients. It means that in the same situations, every 1% change in the LWP will have about eight times more benefits than every 1% change in the other variables.

Fifthly, the local governments of different regions should disseminate policies that align with their economic situation rather than always imitating policies from other regions. For

**Table 15. Threshold value estimation.**

| Model | | Threshold Value | Lower | Upper |
|---|---|---|---|---|
| Model 1 Eastern Region | Th-1 | 18.8000 | 18.7835 | 18.8100 |
| Model 2 Central Region | Th-1 | 16.3210 | 16.3005 | 16.4080 |
|  | Th-21 | 16.3210 | 16.3005 | 16.4080 |
|  | Th-22 | 17.0750 | 17.0310 | 17.0800 |
| Model 3 Western Region | Th-1 | 13.1000 | 12.8815 | 13.1030 |

**Table 16. Threshold regression estimation.**

| Model 1 Eastern Region | | | Model 2 Central Region | | | Model 3 Western Region | | |
|---|---|---|---|---|---|---|---|---|
| Variable | Coefficient | Robust Std.Err | Variable | Coefficient | Robust Std.Err | Variable | Coefficient | Robust Std.Err |
| LEI≤18.80 | 2.738*** (6.37) | 0.430 | LEI≤16.321 | 3.050*** (10.16) | 0.300 | LEI≤13.10 | 1.426*** (2.49) | 0.572 |
| 18.80< LEI | 2.872*** (6.60) | 0.435 | 16.321<LEI≤17.075 | 3.181*** (10.42) | 0.305 | 13.10<LEI | 1.667*** (2.86) | 0.523 |
| | | | 17.075<LEI | 3.290*** (11.05) | 0.298 | | | |
| LIL | -1.032*** (-32.58) | 0.032 | LIL | -0.618*** (-4.85) | 0.127 | LIL | -0.878*** (-6.75) | 0.130 |
| LUE | -0.307 (-1.76) | 0.174 | LUE | -0.107 (-0.40) | 0.270 | LUE | -1.289*** (-4.30) | 0.300 |
| LCDR | 2.418*** (3.21) | 0.752 | LCDR | 2.758*** (3.90) | 0.707 | LCDR | -0.730 (-0.69) | 1.059 |
| LWP | 17.487*** (3.97) | 4.400 | LWP | 17.201*** (5.46) | 3.149 | LWP | 1.160 (0.23) | 5.082 |
| Constant | -77.266*** (-3.50) | 22.081 | Constant | -79.440*** (-4.89) | 16.240 | Constant | 6.974 (0.26) | 26.325 |
| R-square | 0.6927 | | R-square | 0.9180*** | | R-square | 0.4323 | |
| F-test | 696.45*** | | F-test | 2379.13*** | | F-test | 72.92*** | |

Note: Figures in parentheses are T-stat and ***, **, and * show the levels of significance at 1%, 5%, and 10% respectively

instance, the local government of China's central region can expand the sum of export and import to increase the positive impact of the aging population on economic development. However, if the government of China's western region utilizes the same policies, the benefits will relatively be limited. Similarly, introducing policies that can increase the employment rate is more important for China's western region. The above conclusions prove that the aging population has differential impacts on economic growth in different regions of China.

## Policy implication

The research findings indicate that in China, the present aging population has a positive relationship with economic development in all regions. However, with the increasing number of them, eventually, they will have a negative influence on economic development. Therefore, the Chinese government should take precautions and issue forward-looking policies to minimize the negative impacts the aging population might create on future economic development.

Based on the research outcome, this study has several policy suggestions. Firstly, the government should specifically increase the percentage of working people in the eastern and central regions of China. There are a few ways to achieve this goal. For example, increasing the birth rate can essentially solve the problem of the future labor force population. It can also solve the problem of the aging population. The Chinese government has commenced the three-child policy but that has not improved the country's birth rate. Therefore, we can deduce the crucial reason for the decline in the birth rate is that the citizens do not want to have children. Research by Cai and Wang [52] pointed out that the Chinese government should improve welfare policies to increase citizens' willingness to have children. For example, extending maternity leave and providing paternity leave for men. However, this method

**Table 17. Threshold value estimation.**

| Model | | Threshold Value | Lower | Upper |
|---|---|---|---|---|
| Model 4 Eastern Region | Th-1 | 1.0900 | 1.0650 | 1.1310 |
| Model 5 Central Region | Th-1 | 1.3610 | 1.2795 | 1.3860 |
| Model 6 Western Region | Th-1 | 1.1940 | 1.1630 | 1.2240 |
| | Th-21 | 1.1310 | 1.0820 | 1.1630 |
| | Th-22 | 1.3080 | 1.2670 | 1.3350 |

**Table 18. Threshold regression estimation.**

| Eastern Region | | | Central Region | | | Western Region | | |
|---|---|---|---|---|---|---|---|---|
| Variable | Coefficient | Robust Std.Err | Variable | Coefficient | Robust Std.Err | Variable | Coefficient | Robust Std.Err |
| LUE≤1.099 | 1.582*** (6.93) | 0.228 | LUE≤1.361 | 1.267*** (4.18) | 0.303 | LUE≤1.131 | 1.907** (2.96) | 0.645 |
| 1.099<LUE | 1.492*** (6.18) | 0.242 | 1.361<LUE | 1.215*** (3.91) | 0.311 | 1.313<LUE≤1.308 | 1.787** (2.68) | 0.668 |
| | | | | | | 1.308<LUE | 1.684** (2.54) | 0.664 |
| LIL | -0.625*** (-8.87) | 0.070 | LIL | -0.508*** (-4.33) | 0.117 | LIL | -0.496*** (-7.50) | 0.066 |
| LEI | 0.438*** (5.57) | 0.079 | LUE | 0.429*** (16.33) | 0.026 | LUE | 0.294*** (7.77) | 0.038 |
| LCDR | 1.457*** (3.91) | 0.372 | LCDR | 0.315 (0.49) | 0.647 | LCDR | 0.902 (0.68) | 1.333 |
| LWP | 8.394*** (3.78) | 2.221 | LWP | 3.372 (0.98) | 3.443 | LWP | 9.751 (1.54) | 6.325 |
| Constant | -41.000*** (-3.67) | 11.159 | Constant | -14.961 (-0.88) | 17.082 | Constant | -42.452 (-1.30) | 32.573 |
| R-square | 0.6781 | | R-square | 0.9376 | | R-square | 0.4336 | |
| F-test | 394.30*** | | F-test | 642.44*** | | F-test | 249.92*** | |

Note: Figures in parentheses are T-stat and ***, **, and * show the levels of significance at 1%, 5%, and 10% respectively

requires a long-term period to work. The increase in the birth rate will also put further pressure on the currently working people.

Besides that, the Chinese government can promulgate preferential policies to attract foreign workers. For instance, India has a higher birth rate, and its total population will exceed China's population shortly. At present, India has plenty of ready workforce. If the Chinese government can effectively attract young Indians to work in China, it will immediately increase the percentage of working people. Nevertheless, this method also has its drawbacks. This method cannot solely solve the problem of the aging population. Besides, foreign workers' salaries and other welfare benefits will cause higher costs for business enterprises. Apart from that, the Chinese government also can liberalize its immigration policies. If foreign young people emigrate to China, it will increase the working population, and raise the birth rate in the future. However, the migration may lead to cultural conflicts in later years.

Secondly, China's eastern and western regions' governments should increase the sum of imports and exports. For example, the government can encourage local companies to sell their products abroad by implementing policies such as export tax rebates, lower taxes, and exemptions from rent costs. At the same time, the government also can encourage enterprises to import some products that have comparative advantages. This practice will save costs because the enterprises will be able to concentrate on producing products that have absolute advantages to the country's economy and its people.

Thirdly, in terms of the western region of China, the government should increase the opportunity and level of education, reduce the unemployment rate, and attract more business investments. The above three suggestions have a close connection with each other. For example, if there are only a small number of enterprises in the western region of China, the rate of return on education will be relatively lower. It will lead people not to attain higher education. Besides, if there are hardly any business enterprises, it means that competitiveness among the people will be higher. It also will lead to a higher unemployment rate. Alternately, if people only have lower education, business enterprises will not invest in this region. The reason is, that the workforce is not competent for the job.

## Limitation

This study has several limitations: Firstly, public debt plays an essential role in the impact of the aging population on economic growth. However, due to the lack of data, this study cannot

examine the role of public debt. Secondly, spatial effects also play an essential role in the impact of the aging population on economic growth. Therefore, subsequent studies will examine the spatial effects of the aging population. Thirdly, PTR is the piecewise linear regression. In other words, the relationship between X and Y is divided by M. However, this model does not examine the function of M. Thus, in the subsequent research, I will follow the research of Hussain *et al.* [53] and Xuezhou *et al.* [54] to research the mediation of the sum of import and export and the unemployment rate (both are threshold variables of this study). Fourthly, many scholars, such as Zhang *et al.* [55], have proved that green energy has a significant effect on economic development. Besides, the Chinese government announces that China will achieve carbon neutrality in 2060. Apart from that, the aging population will be more serious in China as well. Thus, the subsequent research will focus on the mediation effect of the aging population on green energy and economic development.

## Supporting information

**S1 Dataset. Dataset for eastern, central and western China.**
(XLSX)

**S1 Appendix.**
(DOCX)

## Author Contributions

**Conceptualization:** Nur Syazwani Mazlan.

**Project administration:** Nur Syazwani Mazlan.

**Supervision:** Nur Syazwani Mazlan, Azali Bin Mohamed, Nor Yasmin Binti Mhd Bani.

**Validation:** Nur Syazwani Mazlan, Bufan Liang.

**Writing – original draft:** Yifan Liang.

**Writing – review & editing:** Yifan Liang, Nur Syazwani Mazlan, Bufan Liang.

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
