## [Decision Letter · Decision Letter 0]

21 Nov 2022

PONE-D-22-25089Regional Impact of Aging Population on Economic Development in ChinaPLOS ONE

Dear Dr. MAZLAN,

Thank you for submitting your manuscript to PLOS ONE. After careful consideration, we feel that it has merit but does not fully meet PLOS ONE’s publication criteria as it currently stands. Therefore, we invite you to submit a revised version of the manuscript that addresses the points raised during the review process.

We look forward to receiving your revised manuscript.

Kind regards,

Wajid Khan

Academic Editor

PLOS ONE

Journal Requirements:

Additional Editor Comments:

I have read your manuscript “Regional Impact of Aging Population on Economic Development in China” and I found many positive things in it. However, in my opinion the paper has some shortcomings in regards to text,

Below I have provided numerous remarks on the text:

1. In several instances, I suggest to cite more relevant and recent studies in the introduction and methodology sections.

2. The introduction should be expanded to include importance, uniqueness and contribution to the literature. The description of the contribution needs to be more forensic, needs to be more focused.

3. You need to add limitations of the study.

Reviewers' comments:

Reviewer's Responses to Questions

**Comments to the Author**

1. Is the manuscript technically sound, and do the data support the conclusions?

Reviewer #1: Yes

Reviewer #2: Yes

2. Has the statistical analysis been performed appropriately and rigorously? 

Reviewer #1: No

Reviewer #2: Yes

3. Have the authors made all data underlying the findings in their manuscript fully available?

Reviewer #1: Yes

Reviewer #2: Yes

4. Is the manuscript presented in an intelligible fashion and written in standard English?

Reviewer #1: Yes

Reviewer #2: Yes

5. Review Comments to the Author

Reviewer #1: The paper has some potential but it seems that the method rather than a scientific inquiry dominated the presented research. The presented methodology is rather sophisticated from the statistical point of view but has some serious flaws in the conceptual layer.

The last paragraph of the introduction should present, in a clear and explicit way, the aim of the article and briefly describe the structure of the article.

The literature review should be rewritten into regular sentences. Now it looks like a ppt bullet points pasted into an article.

The authors write about numerous concepts and aspects but without clear reasoning;

The authors should eliminate all ‘bullet-point sentences’ and rewrite those parts into regular sentences.

The review of previous research is very short, and has a form of a mechanistic and technical review only mentioning previous research rather than analysing their results and identifying the research gap. The review should be expanded and also foreign studies should be included

Reviewer #2: It is a good effort. However, following minor changes are recommended before final publication:

1. Please remove the heading “5.1 Major Findings”. If you present conclusions here would be more appropriate.

2. Please check equation numbers after equation 11. Same number i.e., 1 is repeated erroneously

3. The paper contains many grammatical mistakes. It is recommended to get the paper language edited by any authentic native speaker.

4. Although, this journal encourages to publish replication studies but yet you need to highlight the research contributions. I have found the study contributions at the latter parts of literature review section. Please move them to introduction section.

5. Discussion part is not supported by past literature. Add more studies in this section that accept or negate your findings. Please read and cite the following papers carefully:

https://doi.org/10.3389/fenvs.2022.955744

https://doi.org/10.3389/fenvs.2022.841380

https://doi.org/10.1108/SAJBS-05-2020-0150

6. PLOS authors have the option to publish the peer review history of their article (what does this mean?). If published, this will include your full peer review and any attached files.

Reviewer #1: No

Reviewer #2: No

---

## [Author Response · Author response to Decision Letter 0]

9 Feb 2023

Dear Editors and Reviewers:

Thank you for providing us with the reviewers’ evaluation, comments and suggestions related to the paper entitled as above. We have made the corrections and modifications as commented and suggested. Furthermore, we believe those comments and suggestions will help us achieve more outstanding achievements in the future research.

We sincerely hope this manuscript will finally be acceptable to be published in the ‘Plos One’ journal.

Thank you so much for all your help, and we look forward to hearing back from you soon.

Kind regards,

Authors of the manuscript entitled as above

---

## [Decision Letter · Decision Letter 1]

27 Feb 2023

Regional Impact of Aging Population on Economic Development in China

PONE-D-22-25089R1

Dear Nur Syazwani Mazlan,

We’re pleased to inform you that your manuscript has been judged scientifically suitable for publication and will be formally accepted for publication once it meets all outstanding technical requirements.

Kind regards,

Wajid Khan

Academic Editor

PLOS ONE

---

## [Editor Report · Acceptance letter]

2 Mar 2023

PONE-D-22-25089R1 

Regional impact of aging population on economic development in China: evidence from panel threshold regression (PTR) 

Dear Dr. Mazlan:

I'm pleased to inform you that your manuscript has been deemed suitable for publication in PLOS ONE. Congratulations! Your manuscript is now with our production department. 

Kind regards, 

on behalf of

Dr. Wajid Khan 

Academic Editor

PLOS ONE